# Backscattering Estimation of a Tilted Spherical Cap for Different Kinds of Optical Scattering

**Rongkuan Leng** [1,2] , **Zhi Wang** [1,3,*], **Chao Fang** [1,*], **Lei Liu** [1], **Zhiwei Chen** [1,2] **and Xinxu Cui** [1]

1 Changchun Institute of Optics, Fine Mechanics and Physics, Chinese Academy of Sciences, Changchun 130033, China; lengrongkuan@126.com (R.L.); ccliul@163.com (L.L.); chenzhiwei19@mails.ucas.ac.cn (Z.C.); cuixinxu@163.com (X.C.)
2 University of Chinese Academy of Sciences, Beijing 100049, China
3 School of Fundamental Physics and Mathematical Science, Hangzhou Institute for Advanced Study, UCAS, Hangzhou 310024, China
* Correspondence: wz070611@126.com (Z.W.); fangchao@ciomp.ac.cn (C.F.)

**Abstract:** In many optical engineering applications, a spherical cap shaped optical element is widely used such as concave or convex mirrors in reflective optics. Such an element can also tilt around the vertex which corresponds to an off-axis optical design. The optical backscattering of such an optical element sometimes could be important. For example, in the space-based gravitational wave detection, the backscattering of such an element could be superimposed with the local oscillator and limits the sensitivity of the spacecraft. The scattered contributions depend on the scattering property of the mirror surfaces and the geometrical arrangement including the radius of curvature, the tilt and the interval between the scattering source and detector plane. Based on random estimation method, this paper starts from the radiometry, combines these variables and calculates the theoretical amount of back scattered light for both diffuse and superpolished surfaces. The results are compared with analytical and ray tracing solution. The conclusions can be used to further improve the optical design of the telescope or extended to other cases where the backscattered light should be controlled.

**Keywords:** optical scattering; stray light analysis; gravitational waves; Taiji





## 1. Introduction

Light scattering can be considered as an unwanted source that causes image degradation, output reduction, polarization jitter or phase disturbance. For an image system, the optical surface scattering is one of the critical sources that results in stray light. the irradiance of scattered light that received by a detector is related to the scatter properties of the surface and the geometry of the optical system [1]. The scattering properties of an optical surface can be related to imperfections of the surface such as roughness [2,3], contaminations [4,5], subsurface damage [6] etc.al. These imperfections are inevitable even for high quality system and limits the final performance of an optical system.

In real optical systems, the geometrical effect also plays an important role and relies highly on the optical configuration. However, compared with intrinsic scattering, the geometrical effect can be controlled and hence the expected stray light performance of an optical system can be achieved through rational arrangement of the optical elements. For example, an off-axis design, together with the field stop and the Lyot stop, could exhibit a lower level of stray light than an on-axis design [7]. Furthermore, Gary L. Peterson [8] derived the in-field scattered light contribution of a single element in the optical paths and James E. Harvey [9] applies the calculation into a two-mirror system. It shows that the contribution of the stray light of a component is affected by the beam radius. These methods assume that the in-field scattering direction is the same with the direction of the specular rays which is called forward scattering. However, in some special arrangements, the scattering path is different from specular rays where scattering is backward. For example, in LISA

or Taiji gravitational wave (GW) detection program [10,11], the telescope is a four mirror off-axis system. The space-borne telescope delivers the optical signal from the far-end spacecraft to the inner interferometer and transmits the laser beam from the interferometer simultaneously [12]. The backscattered light of transmitted beam from the telescope is an important source of noise for the received signal and limits the sensitivity of detection. Generally, the additional noise is positively correlated with the fraction of transmitted power which is backscattered and recoupled to the receiving beam [13]. Detailed coupling mechanism can be found in Section 2.2. In order to quantify the backscattered light, a common way is to place the virtual collector plane at the exit of the local optical bench. The optical scattering path is different from that of the sequential imaging system which cannot be calculated directly. The tertiary and the quaternary mirror contribute most of the scattered light [12] and superpolished surfaces are necessary. The measurement of the backscattered light together with the optical path stability acquire a lot of attention these years [14–16]. The quaternary mirror, which directly faces the virtual collector, delivers the scattering from the other surfaces and plays an important role to the backscattering contribution. Therefore, in the telescope design phase, it is important to accurately determine the theoretical limit of the scattering performance with different methods. This paper starts from the radiometry and calculates the flux of a tilted spherical cap shaped sample based on the randomly sampling positions. Diffuse, directional scattering and the inhomogeneous light source are all considered. Relevant error analyses are also demonstrated. The parameters for the scattering control include the scattering distribution of the surface, the radius of curvature, tilt angle and the distance between the vertex of the sample and the detector. In the first section, the relevant quantities are introduced. Next, the flux is calculated based on two types of surfaces. An example of backscattered flux calculation is demonstrated based on the quaternary mirror of GW telescope and the results are compared with analytical results and that from a ray tracing software. The method is accurate, flexible and can be applied for direct scattering or illumination system design.

## 2. Theoretical Background

### 2.1. Definitions

The scattering from an optical reflective surface can be described by the bidirectional reflectance distribution function (BRDF) which is the differential radiance ($L_s$) of a certain scattering surface over the differential irradiance $E_i$ of the incident surface [17]:

$$\text{BRDF}(\theta_i, \theta_s, \varphi_s) = \frac{dL_s(\theta_i, \theta_s, \varphi_s)}{dE_i(\theta_i)} \approx \frac{P_s/\Omega_s}{P_i \cos \theta_s} \tag{1}$$

where $P_s, P_i$ is the scattered power and the incident power. $\Omega_s$ is the solid angle of the scattered beam. $\theta_i, \theta_s, \varphi_s$ is the incident angle, scattering angle and azimuth angle. The unit of BRDF is 1/Sr. The relevant geometry is shown in Figure 1. For an isotropic surface, BRDF exhibits rotational symmetry and can be reduced to an in-plane scattering [18].

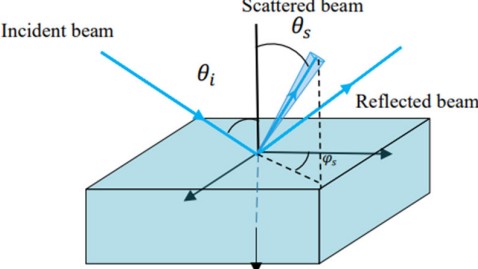

**Figure 1.** Scattering geometry for the definition of BRDF.

The radiometric definition of the scattering is straightforward. However, the accurate measurement and modeling can be difficult. The scattering signature is a result of

the roughness, the surface structure, contaminations, scratches, the wavelength and the polarization. Furthermore, the digital noise, optical alignment also makes the signature even more complicated. In this paper, the scattering of the spherical cap is assumed to be roughness. There are a lot of existing models to simulate the scattering of roughness scattering such as Harvey model [19], K-correction [20], ABG [21] and so on. Detailed comparisons of these models are not the goal of this paper. The model used in this paper is the two-parameter Harvey model.

The total integrate scattering (TIS) is defined as the ratio of total power scattered by a surface in the reflected or transmitted direction to the incident power. For an opaque surface, TIS is defined as:

$$\text{TIS} = \int_0^{2\pi} \int_0^{\pi/2} \text{BRDF} \cdot \sin(\theta_s) \cos(\theta_s) d\theta_s d\varphi_s \tag{2}$$

For high quality isotropic smooth surfaces, the TIS can be used to estimate the root-mean-square roughness which is:

$$\sigma_{\text{rms}} = \frac{\lambda}{4\pi} \sqrt{\text{TIS}} \tag{3}$$

On the other hand, when BRDF is independent from the incident angle or scattering angle, BRDF is a constant and the scattering type is *Lambertian*. Then the ratio of TIS to BRDF is $\pi$. These relations are later used to verify the modeling.

As for the radiation transfer, referring to two randomly oriented surfaces in Figure 2, the differential flux $d^2\Phi$ emitted from $dA_1$ with radiance L and received by $dA_2$ is determined by:

$$d^2\Phi = \frac{L \cdot \cos(\theta_1) \cdot \cos(\theta_2)}{r^2} dA_1 dA_2 \tag{4}$$

where r is the distance between the two infinitesimal surfaces, $\theta_1$, $\theta_2$ is the angle of the scattered ray with respect to the surface normal vector $\vec{n_1}$ and $\vec{n_2}$. When finite surfaces are involved, the total transferred flux can be derived by integration over the total area $A_1$ and $A_2$:

$$\Phi = \int_{A_1} \int_{A_2} \frac{L \cdot \cos(\theta_1) \cdot \cos(\theta_2)}{r^2} dA_1 dA_2 = \int_{A_1} \int_{A_2} \frac{\text{BRDF} \cdot E_{\text{in}} \cos(\theta_1) \cos(\theta_2)}{r^2} dA_1 dA_2 \tag{5}$$

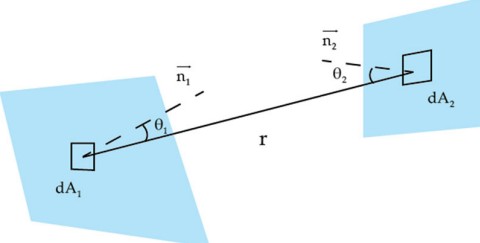

**Figure 2.** Basic geometry for radiation transfer.

As can be seen, the total received flux depends highly on the geometrical layout including the shape, distance, orientation and the area of the surfaces. When the scattering source is *Lambertian*, the radiance term L can be extracted from the integral and the rest divided by the area is often called configuration factor. Some common geometries with different methods can be found in [22]. In the following section, the surface 1 is considered to be a tilted spherical cap and surface 2 is assumed to be a circular planar detector. For simplicity, the view edge effect is not considered which can be found in [23].

When focusing more on the geometrical flux emission characteristics of the emitter itself, the radiance term in Equation (5) can be ignored and the rest is often called Étendue or optical throughput which is defined as [24,25]:

$$\xi = \iint \frac{\cos(\theta_1) \cdot \cos(\theta_2)}{r^2} dA_1 dA_2 \tag{6}$$

Étendue can be used to determine the theoretical flux transfer of an optical system and is often used in non-imaging or illumination design.

### 2.2. Mathmetical Modeling

The configuration of the system is shown in Figure 3. Relevant symbols can be found in Table 1. The optical axis is along Z axis. The detector is a disk with radius $r_1$ on XY plane and the polar coordinate of the infinitesimal patch of the detector is $\rho 2 (0 < \rho 2 < r_2)$, $\varphi_2 (0 < \varphi_2 < 2\pi)$. Then the height of the vertex $z_0$ is determined by:

$$z_0 = d - R + \sqrt{R^2 - {\rho_0}^2} \tag{7}$$

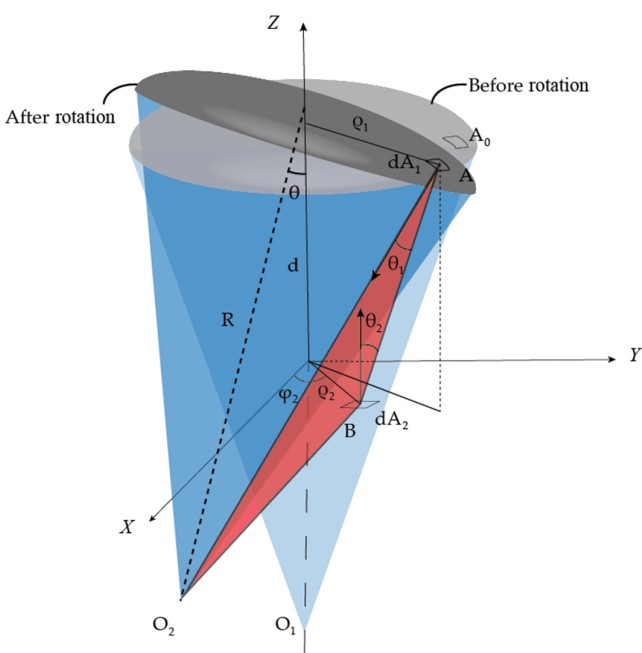

**Figure 3.** Geometrical configuration of a spherical cap segment $dA_0, dA_1$ and an infinitesimal detector surface $dA_2$ at the positions of $A_0$, A and B.

**Table 1.** Relevant Symbols.

| Nomenclature | |
|---|---|
| R | Radius of curvature |
| $\theta$ | Tilt angle |
| d | Distance between the detector and vertex of the spherical cap |
| N | Number of random point pairs |
| $x_{O1}, y_{O1}, z_{O1}$ | The center of spherical cap before rotation |
| $x_{O2}, y_{O2}, z_{O2}$ | The center of spherical cap after rotation |
| $x_0, y_0, z_0$ | Cartesian coordinates of the sample before rotation |
| $\rho_0, \varphi_0, z_0$ | Cylinder coordinates of the sample before rotation |
| $x_1, y_1, z_1$ | Cartesian coordinates of the sample after rotation |
| $x_2, y_2, 0$ | Cartesian coordinates of the detector |
| r | Distance between two differential patches. |
| $A_1, A_2$ | Area of spherical cap and the detector |
| $dA_1, dA_2$ | The differential area of $A_1, A_2$ |

The radiation source is in the shape of a spherical cap and the center, with a height of d radius R, is on the Z axis. The spherical center $O_1$ is $(0, 0, d - R)$. The mirror rotates around X axis with an angle of $\theta$ which is common for an off-axis optical layout. After rotation, the spherical center becomes $O_2$ with the coordinate:

$$\begin{bmatrix} x_{o2} \\ y_{o2} \\ z_{o2} \end{bmatrix} = \begin{bmatrix} 0 \\ d\sin(\theta) - (d - R)\sin(\theta) \\ d - d\cos(\theta) + (d - R)\sin(\theta) \end{bmatrix} \tag{8}$$

For an arbitrary point $A_0$, with the polar coordinate $\rho_0 (0 < \rho_0 < r_1)$, $\varphi_0 (0 < \varphi_0 < 2\pi)$, the point after rotation becomes A and the Cartesian coordinates $x_1, y_1, z_1$ are:

$$\begin{bmatrix} x_1 \\ y_1 \\ z_1 \end{bmatrix} = \begin{bmatrix} \rho_0 \cos\varphi_0 \\ \rho_0 \cos(\theta)\sin(\varphi_0) + d\sin(\theta) - z_0\sin(\theta) \\ d - d\cos(\theta) + z_0\cos(\theta) + \rho_0\sin(\varphi_0)\sin(\theta) \end{bmatrix} \tag{9}$$

Then, the distance r is:

$$r = \sqrt{(x_1 - x_2)^2 + (y_1 - y_2)^2 + z_1{}^2} \tag{10}$$

For an arbitrary point B on the detector, the normal vector is along Z-axis. The normal vector of point A on the spherical cap is along vector $\overrightarrow{AO_2}$. Therefore $\cos(\theta_1)$ and $\cos(\theta_2)$ can be calculated as:

$$\cos(\theta_1) = \frac{\overrightarrow{AO_2} \cdot \overrightarrow{AB}}{\left|\overrightarrow{AO_2}\right|\left|\overrightarrow{AB}\right|}$$
$$\cos(\theta_2) = \frac{z_1}{r} \tag{11}$$

After plugging Equations (7)−(11) together with $dA_1$, $dA_2$ into Equation (5), the total flux emitted from the spherical cap, received by the detector can be derived through the integration. In order to solve the high dimensional integration, a simple feasible method is based on random estimation (Monte Carlo method) where N random pairs of patches $A_i$, $B_i$ are introduced. The basic idea is to randomly sample detector and the spherical cap and the vectors from $A_i$ to $B_i$ are assumed to be the scattered rays. The Cartesian coordinate of $A_i$ is $(x_{1i}, y_{1i}, z_{1i})$ which is a random patch on the spherical cap whereas $B_i$, with the coordinate of $(x_{2i}, y_{2i})$, is a random patch on the detector. Then total transferred flux become:

$$\Phi \approx \frac{A_1 A_2}{N} \sum_1^N \frac{\cos\theta_1 \cos\theta_2}{r^2} = \frac{A_1 A_2}{N} \sum_1^N \mathrm{BRDF}_i \cdot E_{in} \frac{\overrightarrow{A_iO_2} \cdot \overrightarrow{A_iB_i}}{\left|\overrightarrow{A_iO_2}\right|\left|\overrightarrow{A_iB_i}\right|} \frac{z_i}{r^3} \tag{12}$$

The Monte-Carlo method has the following advantages. First, the standard deviation is proportional to $1/\sqrt{N}$ regardless the dimension and smoothness of the integrand [26]. Therefore, it is useful to solve high dimension integrals, or the integrand is not continuous. Furthermore, Equation (12) directly samples the positions of the differential patches which maximizes the collecting efficiency compared with sampling direction aimlessly. The Monte-Carlo method also exhibits a great flexibility especially when a visibility function is introduced. For example, a Circ function can be inserted in Equation (12) which means that the scattered rays are selected by an aperture.

In the following parts, Equation (12) is used to calculate the backscattering of the quaternary mirror of the telescope for the Taiji GW detection program. The space-borne instrument can be divided into two parts. The first part is the optical bench contains an interferometer and a Proof Mass. More details can be found in Ref. [10]. The other part is a four mirror off-axis telescope which transmits the laser from interferometer and received the laser from the far-end telescope simultaneously. The telescope delivers a gaussian laser beam (wavelength $\lambda = 1064$ nm, transmitted power $P_{tot} = 2$ W, beam waist $w = 1.875$ mm)

to the far-end telescope with a 400 mm entrance pupil. The received power for the receiving end is around 700 pw. The phase noise $\phi_{bsc}(t)$ introduced by the backscattered light is [13,27]:

$$\phi_{bsc}(t) = \frac{4\pi}{\lambda}(l_0 + \delta l_{dis}(t)) = \phi_{static} + \delta\phi_{dis}(t) \tag{13}$$

where $l_0$ is the static optical path length while $\delta l_{dis}(t)$ is the displacement noise from the scattering surfaces. $\phi_{static}$ and $\delta\phi_{dis}(t)$ are the corresponding phase. Then the backscattered field $E_{bsc}$ produces an amplitude and phase modulation to the undisturbed interferometer field $E_{in}$:

$$E_{in} + E_{bsc} = E_{in}(1 + \sqrt{f_r\frac{P_{tot}}{P_{in}}}e^{i(\phi_{static}+\phi_{dis}(t))}) \tag{14}$$

where $P_{in}$ is the power inside the interferometer at a reference point for calculation. $f_r$ is the fraction of the transmitted power which is backscattered by the telescope and recoupled with the receiving beam. Since $P_{tot}, P_{in}$ are fixed, $f_r$ in Equation (14) should be kept small to reduce the impact of backscattering. In order to keep the phase noise at picometer level, the fraction $f_r$ is set to at a level of $10^{-10}$ according to the specification of the current Taiji program. Moreover, due to the high coherence of the laser, the impact scattered light can be described by additional phase or amplitude. The additional phase noise can be suppressed by frequency stabilization [28] and then the amplitude noise is dominant [13].

The local optical layout of the tertiary and the quaternary mirror is shown in Figure 4. The scattering from the tertiary mirror is temporarily not considered. The quaternary mirror, which directly faces to the entrance of the optical bench, delivers the scattering from the other surfaces and plays a major part in the backscattering. So, in the design phase, it is particular critical to accurate estimate the scattering contributions of the quaternary mirror. Assuming the telescope is fed by a fundamental mode gaussian beam with total power $P_{tot}$, Beam waist of $w$, the irradiance at a certain position of the mirror is:

$$E_{in}(x_{1i}, y_{1i}) = \frac{2P_{tot}}{\pi w^2}e^{-2(x_{1i}^2+y_{1i}^2)/w^2} \tag{15}$$

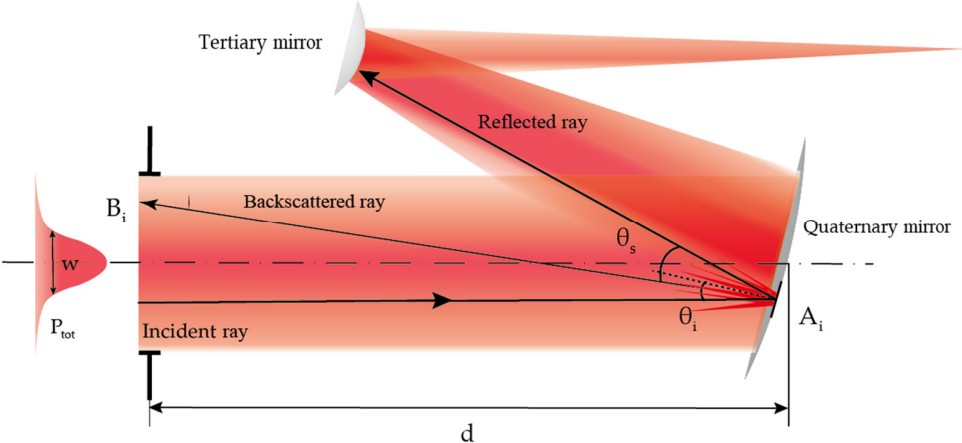

**Figure 4.** Illustration of a backscattered ray of the quaternary mirror which is recoupled to the interferometer.

Plugging Equation (15) into Equation (12), the total flux of the backscattered light can be estimated by:

$$\Phi \approx \frac{A_1 A_2}{N}\sum_{i=1}^{N}BRDF_i \cdot \frac{2P_{tot}}{\pi w^2}e^{-2(x_{1i}^2+y_{1i}^2)/w^2} \cdot \frac{\vec{A_iO_2}\cdot\vec{A_iB_i}}{\left|\vec{A_iO_2}\right|\left|\vec{A_iB_i}\right|}\frac{z_i}{r^3} \tag{16}$$

In Equation (16), $BRDF_i$ is a function of wavelength, incident angle $\theta_i$ and scattering angle $\theta_s$. Since the incident gaussian beam is along Z-axis and the considered scattering is backward, the scattering angle is minus and hence the angle of incidence and the scattering angle fulfill:

$$\cos(\theta_i) = \frac{\overrightarrow{A_iO_2} \cdot \overrightarrow{n_2}}{\left|\overrightarrow{A_iO_2}\right|\left|\overrightarrow{n_2}\right|}$$

$$\cos(\theta_s) = \frac{\overrightarrow{A_iB_i} \cdot \overrightarrow{n_1}}{\left|\overrightarrow{A_iB_i}\right|\left|\overrightarrow{n_1}\right|} \tag{17}$$

## 3. Results

In this section, the results of the scattering flux of a spherical cap are presented. In this paper, two kinds of samples are considered, Section 3.1 demonstrates the results for the diffuse sample under different configurations and the results are compared with existing models. Section 3.2 shows the results of the backscattering of a quaternary mirror of the GW telescope which is superpolished. The scattering model here used is the two-parameter Harvey model. In the model, the parameters of b and s are introduced where b means the BRDF value at $0.573°$ from the direction of specular direction and s means the slope of BRDF on the log-log plot. Section 3.3 demonstrates the relevant error analysis.

### 3.1. Diffuse Samples

A perfect diffuse surface scatters the incident beam equally in all directions meaning that BRDF is independent from incident angle and the scattering angle. For a diffuse and homogeneous incident beam, the scattered radiance is a constant and can be extracted from the integration. Then Equation (5) can be simplified to:

$$\Phi = L \int\limits_{A_1} \int\limits_{A_2} \frac{\cos(\theta_1) \cdot \cos(\theta_2)}{r^2} dA_1 dA_2 = MA_2F \tag{18}$$

where M is the exitance and F is usually called configuration factor, which is:

$$F = \frac{1}{A_2\pi} \int\limits_{A_1} \int\limits_{A_2} \frac{\cos(\theta_1) \cdot \cos(\theta_2)}{r^2} dA_1 dA_2 \tag{19}$$

For such a surface, the geometrical effect is predominant and the configuration factor denotes the fraction of energy emitted or reflected and is collected by the detector. The determination of the configuration factor under different layouts is well-established. Common ways are:

1. Direct analytical method.
2. Statistical method, since the scattering is independent from direction, the configuration factor approximates the number of rays received by the detector divided by the number of rays emitted from the scattering source.
3. Algebraic method, the laws of closeness, reciprocity, distribution and composition can be applied to derive the target configuration factor.
4. Projection method, under some special layouts, the target surface can be projected on a unit sphere in order to simplify the calculation.

For a spherical cap, by considering the tilt angle, shape, size and the distance, a rigorous analytical integral can be developed however it is complicated to solve. With the numerical methods, an approximate solution can be applied to the problem and exhibits considerable freedom for arbitrary arrangements. In the following, the calculated configuration factors at some special cases are compared with the existing analytical solutions. The results of the diffuse spherical cap are shown in Figure 5 and the number of random patch pairs are 1500. In Figure 5a, R = 330 mm, $r_1 = \sqrt{2}/2 \cdot R$, d ranges from $R - \sqrt{R^2 - r_1^2}$ to R,

$\rho_2$ ranges from 1 to $r_1$. When the size of the detector is much smaller than d, the detector can be considered to be infinitesimal and the configuration is calculated though:

$$F = \frac{A_2}{A_1} \frac{R^2 \sin^2 \alpha}{[d - (R - R\cos\alpha)]^2 + R^2 \sin^2 \alpha} \tag{20}$$

where $\sin\alpha = r1/R$ Detailed derivations with Stokes theorem can be found in Appendix A. The result is plotted as the red solid line in Figure 5a. Another interesting case is that d becomes close to $R - \sqrt{R^2 - r_1^2}$. In such a case, the detector plane is on the projection plane of the spherical cap. Since all the radiation emitted from the detector can be collected by spherical cap, the configuration factors can be easily derived with reciprocity relation. The analytical configuration factor for such a case is $A_2/A_1$ and when $\rho_2 = r_1$, the configuration factor is $\cos^2(\alpha/2)$. The results are also plotted as the blue solid line in Figure 5a. The analytical results are in good agreement with the analytical solutions. For larger $r_2$, it shows more relative errors. The reason is straightforward: the standard deviation is inverse proportional to square root of N. Therefore, a larger area needs more sampling pairs which is one of the key parameters for accurate Monte Carlo estimation. More details will be explained in Section 3.3.

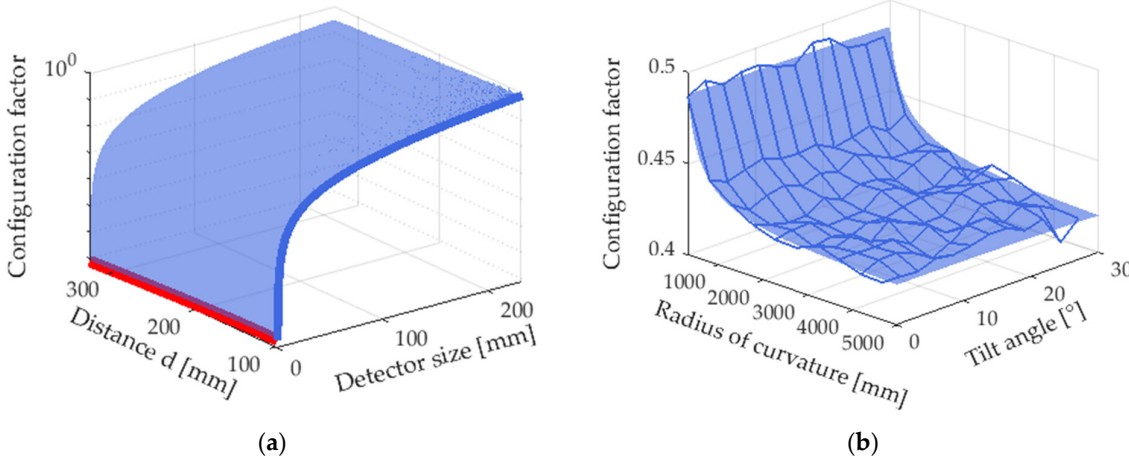

**Figure 5.** The calculated results of the configuration factors under different optical layouts. (**a**) The configuration factors as a function of distance d, detector size $r_2$. The blue and orange solid lines are the analytical results; (**b**) The plot of configuration factors as a function of radius of curvature R and tilt angle. The curve is the analytical results and the grid is for the calculated result.

The other case is the configuration factor changes with the radius of curvature and tilt angle. In the optical layout, r2 = 2.5 mm, d = 118 mm R ranges 330 mm to 5000 mm, $\theta$ from $0°$ to $30°$. The results are shown in Figure 5b where the grid is for the calculated results and the blue curve is for the analytical results. Full analytical expressions can be found in Appendix A. The figure shows that as R increasing, the spherical cap becomes close to a disc and the result for this case fulfills the following well-known relation:

$$F = \frac{1}{2}[X - \sqrt{X^2 - 4\left(\frac{Y_1}{Y_2}\right)^2}] \tag{21}$$

Relevant quantities can be also found in [29]. On the other hand, it can be seen that the scattering from a spherical cap has small dependence on the orientation which meets the assumption of *Lambertian* scattering.

### 3.2. Backscattering of the Quaternary Mirror of Taiji GW Telescope

In this section, a superpolished spherical cap is considered. Compared with the diffuse sample, the most obvious difference is that the scattering is directional. On the other hand, in practice the profile of the incidence irradiance is also not necessary to be uniform. Therefore, the radiance term in Equation (5) has to be kept in the integral and a quite different conclusion can be drawn. This section deals with such a case and in order to calculate the backscattering of the quaternary mirror of Taiji GW telescope. For simplicity, the total power of the incident gaussian beam is set to be unit which is easy to calculate the fraction $f_r$. The diameter of the mirror is 8 mm and the diameter of the collector is 5 mm. The scattering distributions of the mirror are simulated with Harvey model. Figure 6 is the illustration of BRDF under different angle of incidence where b = 0.001, s = −2. With the considered configuration, TIS is 17.9 ppm which corresponds to $\sigma_{rms} \approx 0.36$nm. In practice, these parameters should be modified according to the real measured data. Then, the flux of the backscattered light can be calculated as a function of scattering distribution, tilt angle and radius. The calculated results are compared with commercial soft ASAP [30] which is famous for the outstanding capability, flexibility, speed and accuracy. The scattering calculation of ASAP is based on ray tracing which additionally generates a set of scattering direction cosines in the importance edges and the energy is not always conserved [31].

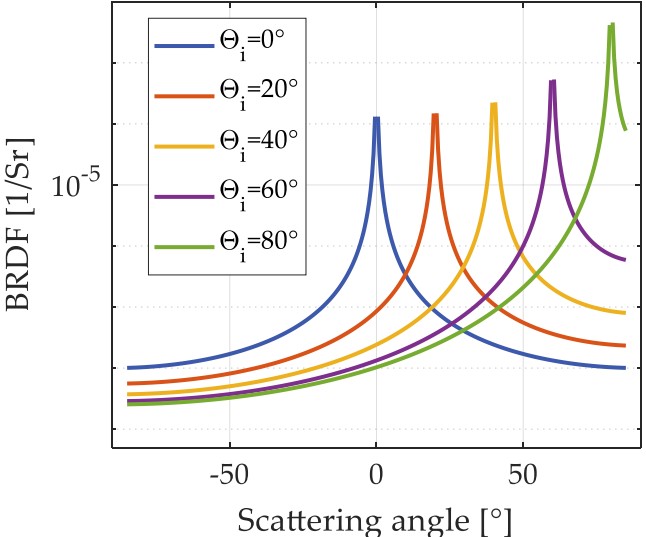

**Figure 6.** Scattering distribution simulation based on two-parameter Harvey model.

The calculated backscattering flux changes with distance d is shown in Figure 7a where the mirror has tilt angle of 5°. For Monte-Carlo estimation, 10,000 random pairs of patches are chosen and the backscattered flux is plotted the blue curve and the gray curve is the corresponding standard deviation. The results are in good agreement with ASAP calculation (Figure 7a red curve) for large distance. The backscattering flux of Monte-Carlo methods at d = 118 mm is $3.09 \times 10^{-10}$W whereas the ASAP result is $4.5134 \times 10^{-10}$W. The flux increases dramatically as the distance decreases which is because the detector goes into the specular reflected paths which can be seen in Figure 4. On the other hand, for small distance, a larger variance can be witnessed which can be explained by the geometrical effects. As mentioned, BRDF is a strong function of angle of incidence and scattering angle. When the distance is small, the normal vectors of the spherical cap different positions are different meaning that the local patches scatter the incident beam differently. However, as the distance increases and when the area of the spherical cap is much smaller than the distance, the geometrical shape of the scattering source has little effect on scattering. This assumption is commonly used in remote sensing. In some literature, the geometrical effect is simplified to the solid angle when d is much larger than the size of scattering source. Figure 7b shows the evolvement of the average of incidence angle and the scattering angle

from small distance to large distance. The error bars represent the standard deviation of the incident and scattering angles at different positions. For the considered case, the angle of incidence (Blue curve in Figure 7b) is near a constant of 5° which corresponds to the tilt angle of the spherical cap. The scattering angle (Red curve in Figure 7b) are minus meaning the backward scattering and both the average and the variance for small distance are larger than that of larger distance.

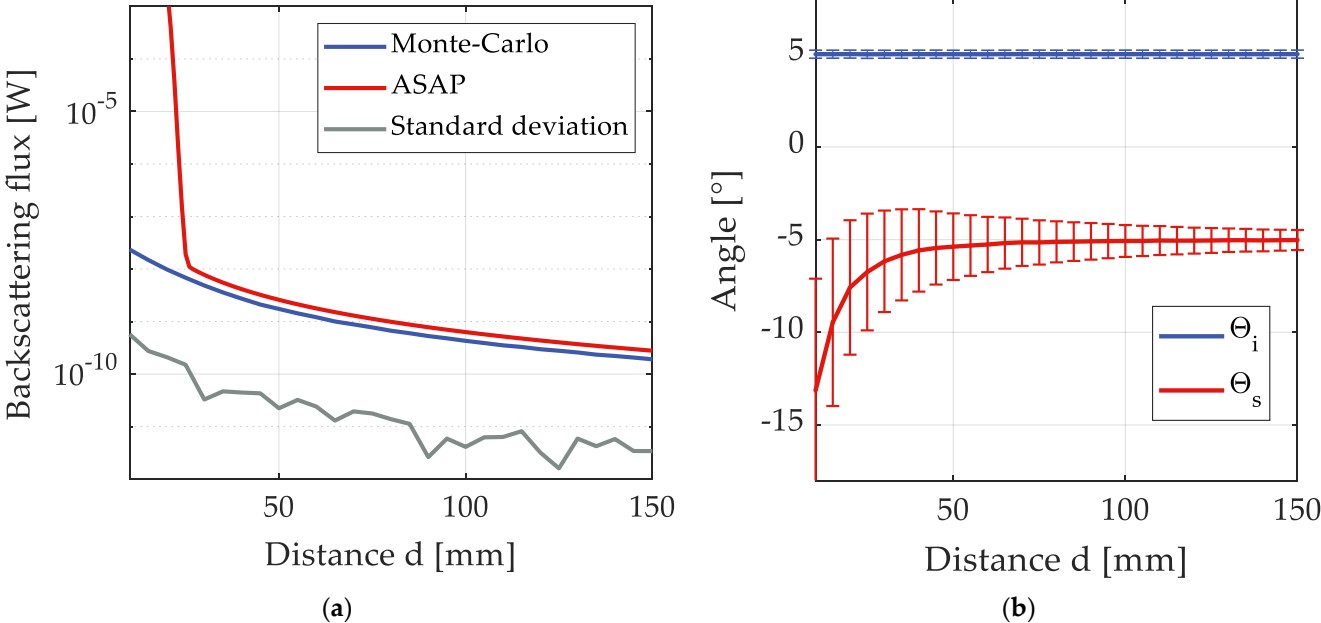

**Figure 7.** The backscattering behavior of a superpolished spherical cap. (**a**) The backscattering as a function of distance d. the gray curve is standard deviation of Monte-Carlo estimation. (**b**) the angle of incidence and scattering angle with respect to the distance and the corresponding standard deviation for different position.

Furthermore, the standard deviation of scattering angle has the same tendency of the standard deviation of the final scattering flux which also explains the reason why large distance calculation is stabler and more accurate. So, threshold distance $d_{threshold}$ is the position from which the geometrical effect becomes stable. It defined as the distance where the standard deviation product of the scattering angle $\sigma_i(d)$ and incident angle $\sigma_s(d)$ equals to $1/e$ of the maximum. In detail, the distance $d_{threshold}$ fulfills:

$$\sigma(d_{threshold}) = \frac{\max[\sigma_s(d) \cdot \sigma_i(d)]}{e} \qquad (22)$$

In Equation (22), the standard deviation of angle of incidence and scattering angle are chosen rather than that of the final backscattering which can minimize the impact of the numerical calculation. For the considered configuration, $d_{threshold}$ is calculated as 34 mm.

Another important parameter for backscattering control is the tilt angle of the spherical cap. As mentioned, the reflection becomes directional and as a result, the backscattering performance of a superpolished surface relies highly on the tilt angle and hence the performance of backscattering can be controlled. This fact is illustrated in Figure 8. where the backscattering flux is calculated as a function of tilt angle and the roughness of the mirror. The blue curves are from the Monte-Carlo method and the red curves are for ASAP. The solid lines are for the calculations whose RMS = 0.36 nm, the dashed lines are for the surface with 0.6 nm and the dotted lines are for RMS = 1.42 nm. The backscattering flux increases as the roughness increases. The Monte-Carlo method and the ASAP calculation show the same tendency of decreasing of the total flux with the increasing of the tilt angle. From the graph, one orders drop of backscattering flux can be seen as the tilt angle from 5°

to 10°. A larger tilt angle usually means a larger scattering angle. As mentioned before, the scattering angle with respect to the specular ray is approximately equals to twice of the tilt angle for a large distance and radius of curvature. Therefore, the received flux can be controlled by properly arranging the tilt angle of the mirror.

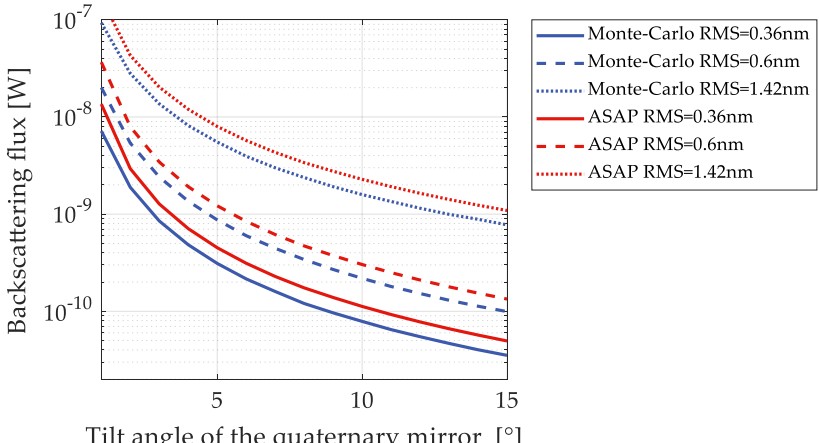

**Figure 8.** Backscattering flux as a function of tilt angle for reflective spherical cap with different roughness. The blue curves are the calculation with Monte-Carlo method and the red curves are for ASAP results.

On the other hand, it is also found that when the optical layout fulfills Equation (22), the standard deviation of the incident angle and the scattering angle are small meaning that the ratio of backscattered flux to BRDF is around a constant which equals to:

$$\frac{\Phi}{BRDF} \approx \int\limits_{A_1}\int\limits_{A_2} E_{in}(x,y) \cdot \frac{\cos(\theta_1) \cdot \cos(\theta_2)}{r^2} dA_1 dA_2 \tag{23}$$

Figure 9 shows the ratio of backscattered flux to BRDF under different roughness, As can be seen, the ratio is a constant around $9.5143 \times 10^{-4} W \cdot Sr$ and is independent from the roughness of the surface. Moreover, when the incident irradiance is homogeneous, Equation (23) can be further related to Étendue and the scattered flux can be easily calculated with:

$$\Phi = BRDF \cdot E_i \cdot \xi \tag{24}$$

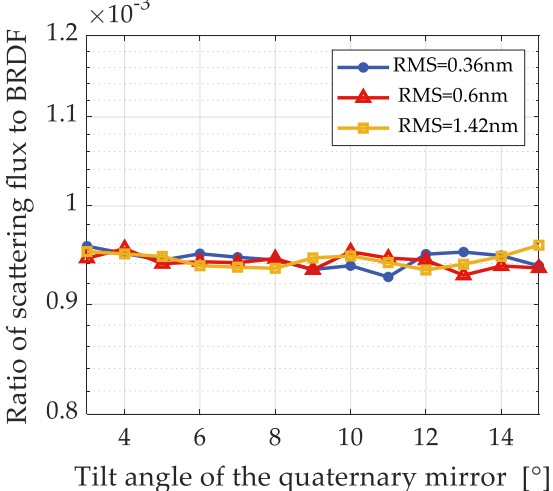

**Figure 9.** The ratio of backscattering flux to BRDF with respect to the tilt angle for the samples under different roughness.

*3.3. Error Analysis*

The quality of the integral can be increased with the following strategies. The first and the simplest one is to increase the number of samplings. As mentioned before, the standard deviation is inverse proportional to $\sqrt{N}$ which makes Monte-Carlo to be a practical technique for high dimension problems. However, the square root of the number of samplings may lead to low efficiency and in practice there always exists a convergence rate issue even though the N is large.

Importance sampling reduces variance by changing the probability density function. In Equation (12), the probability density function is set to be uniform which is $A_1A_2$. Therefore, variance always exist because of the mismatch between the probability density function and the integrated function. An ideal case is to find a probability density function that is proportional to the function and the variance is zero. However, such a distribution is difficult to find. Since the value of the integral must be known beforehand. A common method is to estimate the probability density function according to a certain known term is of the integrand [26]. Since the function is smooth and continuous for the considered optical layout, the improvement is limited and the probability density function is kept uniform. However, the quality of the randomness is critical and directly affects the accuracy of the results. A pseudorandom process appears to be random but is not. It will cause the mismatch between the samplings and the probability density function in Equation (12) and is fatal to the results. More details about such an effect can be found in [32,33]. Modern methods to generate random points include: N-Rooks [34], Multi-jittered [35] which are aiming at acquiring better randomness during the sampling.

The other practical technique is to control the source of variance. Since the sampling patches are independent, by controlling the source of variance can reduce the overall variance. In this paper, controlling the rays with extreme value is an effective method. After verification, the largest source of variance in Equation (12) is the BRDF term especially when near the specular direction. These rays should be less weighted or resampled in the iteration in order to increase the robustness and the accuracy of the algorithm. More details can be found in the source code.

## 4. Discussion

In this paper, a method of calculating the direct scattering flux is raised which is not only suitable for diffuse surfaces but also for the case where the incident irradiance and the scattering distribution of the scattering source are not uniform. The calculation starts from radiometry with the method of random estimation. The basic idea is to choose a larger number of random patch pairs between the scattering surface and detector surface and calculate the radiation transfer between these patches. In this way, the collecting efficiency can be maximized. It is also demonstrated that how the integral of flux calculation is simplified under different situations. The result can be applied to heat transfer, optical scattering or illumination calculation. For diffuse surfaces, the configuration factors are calculated and are in good agreement with the analytical solutions under some specific configurations. As for the scattering flux for a superpolished surface, the theoretical best backscattering performance of the quaternary mirror of off-axis telescope in space GW detection is estimated where the incident beam is gaussian. The results show that the distance between the spherical cap and the detector, tilt angle, and the scattering property are the key parameters in scattering control. With the current design, it is recommended that the root mean square roughness smaller than 0.4 nm and the tilt angle larger than $5°$. It is worth mentioning that these conclusions only consider the stray light effects. For the final optical design, more effects should be taken into consideration and therefore, a compromised value should be chosen. For example, a larger tilt angle usually introduces more off-axis aberration such as coma. However, the results could be a reference in the optical design phase. The relevant error analyses are also given in this paper. It shows that the randomness of the sampling and the controlling of the extreme values are crucial for the algorithm. The random estimation method exhibits great flexibility and accuracy for

high dimension problems. By following the structure of the paper, the scattering flux of other geometrical layouts could also be calculated. In addition, based on the approach, the problem of radiation transportation could be further solved which is indirect scattering or multi-scattering. Moreover, with random estimation, more information can be integrated such as the visibility or polarization reaction of the surface.

**Author Contributions:** Conceptualization, R.L.; methodology, R.L.; software, R.L. and Z.C.; validation, R.L. and Z.C.; formal analysis, R.L.; investigation, R.L.; writing—original draft preparation, R.L.; writing—review and editing, Z.W. and C.F.; visualization, X.C.; resources, X.C.; supervision, Z.W., C.F. and L.L.; funding acquisition, Z.W. and C.F. All authors have read and agreed to the published version of the manuscript.

**Funding:** This research was funded by National Natural Science Foundation of China, grant number 62075214, National Key R&D Program of China, grant number 2020YFC2200104.

**Data Availability Statement:** The data and the source code are publicly available on "https://github.com/LENGRK/LRK" (accessed on 10 March 2022).

**Conflicts of Interest:** The authors declare no conflict of interest.

## Appendix A

The configuration factor of a tilted spherical cap can be analytically derived as follows. According to the Stokes theorem, the configuration factor can be expressed as:

$$F = n_1 \int \frac{y_1 dx_1 - x_1 dy_1}{2\pi r^2} \tag{A1}$$

Plugging Equations (7)–(11) into (A1), the result is:

$$F = \frac{A_2}{A_1} \int_0^{2\pi} \frac{C + D \cdot \sin \varphi_0}{2\pi(A + B \sin \varphi_0)} d\varphi_0 \tag{A2}$$

The parameter used are defined as follows:

$$
\begin{aligned}
A &= d^2 + 2R^2 - r1^2 - 2R\sqrt{R^2 - r1^2} + \rho_0{}^2 + 2d\left(-R + \sqrt{R^2 - r_1{}^2}\right)\cos\theta \\
B &= 2dr_1 \sin\theta \\
C &= -r_1{}^2 \cos\theta \\
D &= r_1(\sqrt{R^2 - r_1{}^2} - R)\sin\theta
\end{aligned}
\tag{A3}
$$

When $\theta = 0$, B and D become 0 and Equation (A2) can be simplified into Equation (20).

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
