# Peer review of "Backscattering Estimation of a Tilted Spherical Cap for Different Kinds of Optical Scattering"

_optics, doi:10.3390/opt3020018_

Round 1
Reviewer 1 Report
- Please, could you highlight your contributions more clearly in the Introduction section? It is crucial for readers.
- Section 3.1 demonstrates the results for the diffuse sample under different configurations and the results compared with existing models. It is necessary to emphasize these models in this place!
- The following paragraph is in Section 3.3. (Error analysis), and should be important to reformulate in such a way as to add appropriate clarification: "Therefore, variance always exists because of the mismatch between the probability density function and the function being integrated. An ideal case is to find a probability density function that is proportional to the function and the variance is zero. However, such a distribution is difficult to find. " Please proceed and explain why?
- It is necessary to give much more details in the Conclusion.
Reviewer 2 Report
The paper "Backscattering Estimation of a Tilted Spherical Cap for Different Kinds of Optical Scattering" reports method of calculating the backscattering flux of a tilted spherical cap is raised which is not only suitable for diffuse surface but also for the case where the incident irradiance and the scattering distribution of the scattering source are not uniform. The basic theoretical propositions are recognized in the scientific community and are beyond doubt. The title of the article reflects its content. The abstract is concise and presents the main theses of the work. The conclusions and the correlation analysis conducted are undeniable. The structure of the paper is concise and the scientific level is high.
In summary - I will recommend for publication this article in the Optics.
Author Response
Dear reviewer:
Thanks for your patience and I really appreciate your comments.
Best wishes
Reviewer 3 Report
The manuscript of R. Leng et al. is devoted to the modelling of light backscattering in interferometers for gravitational wave observation. The topic is novel and interesting, but the specific problem, addressed by the authors is not sufficiently new. The used methods are also quite common.
The main question. If the authors have access to ASAP software and know how to use it, why not use it for interferometer design? What was the motive for implementing simplistic Monte-Carlo models (with all their potential pitfalls)?
Maybe, the poor language quality and structuring of the manuscript prevented the accurate estimation of its scientific value. But, in our opinion the manuscript cannot be accepted in its present form.
Please do not include any copyrighted work (articles, theses), which is not yours, to the Supplementary materials .
Round 2
Reviewer 3 Report
Most of the questions concerning application and novelty has been cleared by the authors’ response. It became clear which part of the whole interferometric system is considered, and why the authors need to develop new computational tools. If some of the important details from the response were included in the manuscript, it would be better. But, unfortunately, they were not included even in the new version.
Please find below a list of some minor points and some language issues (Please note, that (a) I am not a native English speaker, (b) the list may not be complete, and (c) it is not a reviewer’s duty to check grammar, so I strongly recommend to consult a native speaker)
Line 72, 306 and later in the text ‘tilted angle’ (?). Maybe ‘tilt angle’?
Line 73 ‘luminous problem’ – confusing
Line 97 It should probably be ‘complicated’ instead of ‘complicate’
Lines 98-99 – very confusing
Figure 4. Please provide the labels for the tertiary and quaternary mirror for clarity. It will be also beneficial to place the label ‘Backscattered ray’ above to the ray (now it actually names the dash-dot line/optical axis)
Line 166 - ‘Besides, Eq. (12) direct samples the positions’ – this part is unclear. Maybe ‘directly samples’, but still not sure
Line 163, 304, 309-314, 343. ‘Standard variance’ is an incorrect term. Either ‘variance’ σ2, which is a distribution parameter, or ‘standard deviation’ or ‘RMSD’ S, which is a statistic. Please check carefully which case is which.
Line 173 – grammar issue
Line 181 – picometer is unit of length (1E-12 meter). Probably you have meant ‘picowatt meter’?
Line 189 – ‘to the entrance of the entrance of the optical bench’ – word redundancy
Line 193 – very confusing
Lines 202-203 – confising. Maybe ‘existing models’?
Line 207 – ‘value at 0.573° from specular’ (?) Specular is an adjective. Maybe ‘from the direction of specular reflection’?
Line 235 – ‘existed models’ --> ‘existing models’ (?)
Line 256 – what does 'theoretical' mean here? Does it mean ‘analytical’? You do not do actual optical measurements, so all the results are actually theoretical.
Line 257 – ‘closed to a disc’ --> ‘becomes close to a disc’
Line 269 – ‘On the hand’ (?)
Lines 277-278 – why and how these specific values for b and s were selected?
Line 294 - ‘gets into the paths of specular reflected paths’ – word redundancy
Figure 7, vertical axis title. ‘angle of degree’ (?) Maybe ‘angle’
The discussion in lines 359-361 and figure 10 are excessive. Just a small note and reference to work [34] is enough. There are tons of examples on this thing, fast uniform sampling on the N-dimensional sphere is one of the most well-known and common Monte-Carlo subproblems.
